# GP Kernels for Cross-Spectrum Analysis

[1]**Kyle Ulrich,** [3]**David E. Carlson,** [2]**Kafui Dzirasa,** [1]**Lawrence Carin**
[1]Department of Electrical and Computer Engineering, Duke University
[2]Department of Psychiatry and Behavioral Sciences, Duke University
[3]Department of Statistics, Columbia University
{kyle.ulrich, kafui.dzirasa, lcarin}@duke.edu
david.edwin.carlson@gmail.com

## Abstract

Multi-output Gaussian processes provide a convenient framework for multi-task problems. An illustrative and motivating example of a multi-task problem is multi-region electrophysiological time-series data, where experimentalists are interested in both power and phase coherence between channels. Recently, Wilson and Adams (2013) proposed the spectral mixture (SM) kernel to model the spectral density of a single task in a Gaussian process framework. In this paper, we develop a novel covariance kernel for multiple outputs, called the cross-spectral mixture (CSM) kernel. This new, flexible kernel represents both the power and phase relationship between multiple observation channels. We demonstrate the expressive capabilities of the CSM kernel through implementation of a Bayesian hidden Markov model, where the emission distribution is a multi-output Gaussian process with a CSM covariance kernel. Results are presented for measured multi-region electrophysiological data.

## 1 Introduction

Gaussian process (GP) models have become an important component of the machine learning literature. They have provided a basis for non-linear multivariate regression and classification tasks, and have enjoyed much success in a wide variety of applications [16].

A GP places a prior distribution over latent functions, rather than model parameters. In the sense that these functions are defined for any number of sample points and sample positions, as well as any general functional form, GPs are nonparametric. The properties of the latent functions are defined by a positive definite covariance kernel that controls the covariance between the function at any two sample points. Recently, the spectral mixture (SM) kernel was proposed by Wilson and Adams [24] to model a spectral density with a scale-location mixture of Gaussians. This flexible and interpretable class of kernels is capable of recovering any composition of stationary kernels [27, 9, 13]. The SM kernel has been used for GP regression of a scalar output (i.e., single function, or observation "task"), achieving impressive results in extrapolating atmospheric $CO_2$ concentrations [24]; image inpainting [25]; and feature extraction from electrophysiological signals [21].

However, the SM kernel is not defined for multiple outputs (multiple correlated functions). Multi-output GPs intersect with the field of multi-task learning [4], where solving similar problems jointly allows for the transfer of statistical strength between problems, improving learning performance when compared to learning all tasks individually. In this paper, we consider neuroscience applications where low-frequency ($< 200$ $Hz$) extracellular potentials are simultaneously recorded from implanted electrodes in multiple brain regions of a mouse [6]. These signals are known as local field potentials (LFPs) and are often highly correlated between channels. Inferring and understanding that interdependence is biologically significant.

A multi-output GP can be thought of as a standard GP (all observations are jointly normal) where the covariance kernel is a function of both the input space and the output space (see [2] and references therein for a comprehensive review); here "input space" means the points at which the functions are sampled (e.g., time), and the "output space" may correspond to different brain regions. A particular positive definite form of this multi-output covariance kernel is the *sum of separable* (SoS) kernels, or the *linear model of coregionalization* (LMC) in the geostatistics literature [10], where a separable kernel is represented by the product of separate kernels for the input and output spaces.

While extending the SM kernel to the multi-output setting via the LMC framework (i.e., the SM-LMC kernel) provides a powerful modeling framework, the SM-LMC kernel does not intuitively represent the data. Specifically, the SM-LMC kernel encodes the cross-amplitude spectrum (square root of the cross power spectral density) between every pair of channels, but provides no cross-phase information. Together, the cross-amplitude and cross-phase spectra form the *cross-spectrum*, defined as the Fourier transform of the cross-covariance between the pair of channels.

Motivated by the desire to encode the full cross-spectra into the covariance kernel, we design a novel kernel termed the *cross-spectral mixture* (CSM) kernel, which provides an intuitive representation of the power *and* phase dependencies between multiple outputs. The need for embedding the full cross-spectrum into the covariance kernel is illustrated by a recent surge in neuroscience research discovering that LFP interdependencies between regions exhibit phase synchrony patterns that are dependent on frequency band [11, 17, 18].

The remainder of the paper is organized as follows. Section 2 provides a summary of GP regression models for vector-valued data, and Section 3 introduces the SM, SM-LMC, and novel CSM covariance kernels. In Section 4, the CSM kernel is incorporated in a Bayesian hidden Markov model (HMM) [14] with a GP emission distribution as a demonstration of its utility in hierarchical modeling. Section 5 provides details on inverting the Bayesian HMM with variational inference, as well as details on a fast, novel GP fitting process that approximates the CSM kernel by its representation in the spectral domain. Section 6 analyzes the performance of this approximation and presents results for the CSM kernel in the neuroscience application, considering measured multi-region LFP data from the brain of a mouse. We conclude in Section 7 by discussing how this novel kernel can trivially be extended to any time-series application where GPs and the cross-spectrum are of interest.

## 2 Review of Multi-Output Gaussian Process Regression

A multi-output regression task estimates samples from $C$ output channels, $\boldsymbol{y}_n = [y_{n1}, \ldots, y_{nC}]^T$ corresponding to the $n$-th input point $x_n$ (e.g., the $n$-th temporal sample). An unobserved latent function $\boldsymbol{f}(x) = [f_1(x), \ldots, f_C(x)]^T$ is responsible for generating the observations, such that $\boldsymbol{y}_n \sim \mathcal{N}(\boldsymbol{f}(x_n), \boldsymbol{H}^{-1})$, where $\boldsymbol{H} = \text{diag}(\eta_1, \ldots, \eta_C)$ is the precision of additive Gaussian noise.

A GP prior on the latent function is formalized by $\boldsymbol{f}(x) \sim \mathcal{GP}(\boldsymbol{m}(x), \boldsymbol{K}(x, x'))$ for arbitrary input $x$, where the mean function $\boldsymbol{m}(x) \in \mathbb{R}^C$ is set to equal $\boldsymbol{0}$ without loss of generality, and the covariance function $(\boldsymbol{K}(x, x'))_{c,c'} = k^{c,c'}(x, x') = \text{cov}(f_c(x), f_{c'}(x'))$ creates dependencies between observations at input points $x$ and $x'$, as observed on channels $c$ and $c'$. In general, the input space $x$ could be vector valued, but for simplicity we here assume it to be scalar, consistent with our motivating neuroscience application in which $x$ corresponds to time.

A convenient representation for multi-output kernel functions is to separate the kernel into the product of a kernel for the input space and a kernel for the interactions between the outputs. This is known as a separable kernel. A *sum of separable kernels* (SoS) representation [2] is given by

$$k^{c,c'}(x, x') = \sum_{q=1}^{Q} b_q(c, c') k_q(x, x'), \qquad \text{or} \qquad \boldsymbol{K}(x, x') = \sum_{q=1}^{Q} \boldsymbol{B}_q k_q(x, x'), \qquad (1)$$

where $k_q(x, x')$ is the input space kernel for component $q$, $b_q(c, c')$ is the $q$-th output interaction kernel, and $\boldsymbol{B}_q \in \mathbb{R}^{C \times C}$ is a positive semi-definite output kernel matrix. Note that we have a discrete set of $C$ output spaces, $c \in \{1, \ldots, C\}$, where the input space $x$ is continuous, and discretely sampled arbitrarily in experiments. The SoS formulation is also known as the *linear model of coregionalization* (LMC) [10] and $\boldsymbol{B}_q$ is termed the coregionalization matrix. When $Q = 1$, the LMC reduces to the *intrinsic coregionalization model* (ICM) [2], and when $\text{rank}(\boldsymbol{B}_q)$ is restricted to equal 1, the LMC reduces to the *semiparametric latent factor model* (SLFM) [19].

Any finite number of latent functional evaluations $\boldsymbol{f} = [f_1(\boldsymbol{x}), \ldots, f_C(\boldsymbol{x})]^T$ at locations $\boldsymbol{x} = [x_1, \ldots, x_N]^T$ has a multivariate normal distribution $\mathcal{N}(\boldsymbol{f}; \boldsymbol{0}, \boldsymbol{K})$, such that $\boldsymbol{K}$ is formed through the block partitioning

$$\boldsymbol{K} = \begin{bmatrix} k^{1,1}(\boldsymbol{x}, \boldsymbol{x}) & \cdots & k^{1,C}(\boldsymbol{x}, \boldsymbol{x}) \\ \vdots & \ddots & \vdots \\ k^{C,1}(\boldsymbol{x}, \boldsymbol{x}) & \cdots & k^{C,C}(\boldsymbol{x}, \boldsymbol{x}) \end{bmatrix} = \sum_{q=1}^{Q} \boldsymbol{B}_q \otimes k_q(\boldsymbol{x}, \boldsymbol{x}), \tag{2}$$

where each $k^{c,c'}(\boldsymbol{x}, \boldsymbol{x})$ is an $N \times N$ matrix and $\otimes$ symbolizes the Kronecker product.

A vector-valued dataset consists of observations $\boldsymbol{y} = \text{vec}([\boldsymbol{y}_1, \ldots, \boldsymbol{y}_N]^T) \in \mathbb{R}^{CN}$ at the respective locations $\boldsymbol{x} = [x_1, \ldots, x_N]^T$ such that the first $N$ elements of $\boldsymbol{y}$ are from channel 1 up to the last $N$ elements belonging to channel $C$. Since both the likelihood $p(\boldsymbol{y}|\boldsymbol{f}, \boldsymbol{x})$ and distribution over latent functions $p(\boldsymbol{f}|\boldsymbol{x})$ are Gaussian, the marginal likelihood is conveniently represented by

$$p(\boldsymbol{y}|\boldsymbol{x}) = \int p(\boldsymbol{y}|\boldsymbol{f}, \boldsymbol{x}) p(\boldsymbol{f}|\boldsymbol{x}) d\boldsymbol{f} = \mathcal{N}(\boldsymbol{0}, \boldsymbol{\Gamma}), \qquad \boldsymbol{\Gamma} = \boldsymbol{K} + \boldsymbol{H}^{-1} \otimes \boldsymbol{I}_N, \tag{3}$$

where all possible functions $\boldsymbol{f}$ have been marginalized out.

Each input-space covariance kernel is defined by a set of hyperparameters, $\boldsymbol{\theta}$. This conditioning was removed for notational simplicity, but will henceforth be included in the notation. For example, if the squared exponential kernel is used, then $k_{\text{SE}}(x, x'; \boldsymbol{\theta}) = \exp(-\frac{1}{2}||x - x'||^2/\ell^2)$, defined by a single hyperparameter $\boldsymbol{\theta} = \{\ell\}$. To fit a GP to the dataset, the hyperparameters are typically chosen to maximize the marginal likelihood in (3) via gradient ascent.

# 3 Expressive Kernels in the Spectral Domain

This section first introduces the spectral mixture (SM) kernel [24] as well as a multi-output extension of the SM kernel within the LMC framework. While the SM-LMC model is capable of representing complex spectral relationships between channels, it does not intuitively model the cross-phase spectrum between channels. We propose a novel kernel known as the *cross-spectral mixture* (CSM) kernel that provides both the cross-amplitude and cross-phase spectra of multi-channel observations. Detailed derivations of each of these kernels is found in the Supplemental Material.

## 3.1 The Spectral Mixture Kernel

A spectral Gaussian (SG) kernel is defined by an amplitude spectrum with a single Gaussian distribution reflected about the origin,

$$S_{\text{SG}}(\omega; \boldsymbol{\theta}) = \frac{1}{2} \left[ \mathcal{N}(\omega; -\mu, \nu) + \mathcal{N}(\omega; \mu, \nu) \right], \tag{4}$$

where $\boldsymbol{\theta} = \{\mu, \nu\}$ are the kernel hyperparameters, $\mu$ represents the peak frequency, and the variance $\nu$ is a scale parameter that controls the spread of the spectrum around $\mu$. This spectrum is a function of angular frequency. The Fourier transform of (4) results in the stationary, positive definite auto-covariance function

$$k_{\text{SG}}(\tau; \boldsymbol{\theta}) = \exp(-\frac{1}{2}\nu\tau^2) \cos(\mu\tau), \tag{5}$$

where stationarity implies dependence on input domain differences $k(\tau; \boldsymbol{\theta}) = k(x, x'; \boldsymbol{\theta})$ with $\tau = x - x'$. The SG kernel may also be derived by considering a latent signal $f(x) = \sqrt{2}\cos(\omega(x + \phi))$ with frequency uncertainty $\omega \sim \mathcal{N}(\mu, \nu)$ and phase offset $\omega\phi$. The kernel is the auto-covariance function for $f(x)$, such that $k_{\text{SG}}(\tau; \boldsymbol{\theta}) = \text{cov}(f(x), f(x+\tau))$. When computing the auto-covariance, the frequency $\omega$ is marginalized out, providing the kernel in (5) that includes all frequencies in the spectral domain with probability 1.

A weighted, linear combination of SG kernels gives the spectral mixture (SM) kernel [24],

$$k_{\text{SM}}(\tau; \boldsymbol{\theta}) = \sum_{q=1}^{Q} a_q k_{\text{SG}}(\tau; \boldsymbol{\theta}_q), \qquad S_{\text{SM}}(\omega; \boldsymbol{\theta}) = \sum_{q=1}^{Q} a_q S_{\text{SG}}(\omega; \boldsymbol{\theta}_q), \tag{6}$$

where $\boldsymbol{\theta}_q = \{a_q, \nu_q, \mu_q\}$ and $\boldsymbol{\theta} = \{\boldsymbol{\theta}_q\}$ has $3Q$ degrees of freedom. The SM kernel may be derived as the Fourier transform of the spectral density $S_{\text{SM}}(\omega; \boldsymbol{\theta})$ or as the auto-covariance of latent functions $f(x) = \sum_{q=1}^{Q} \sqrt{2a_q} \cos(\omega_q(x + \phi_q))$ with uncertainty in angular frequency $\omega_q \sim \mathcal{N}(\mu_q, \nu_q)$.

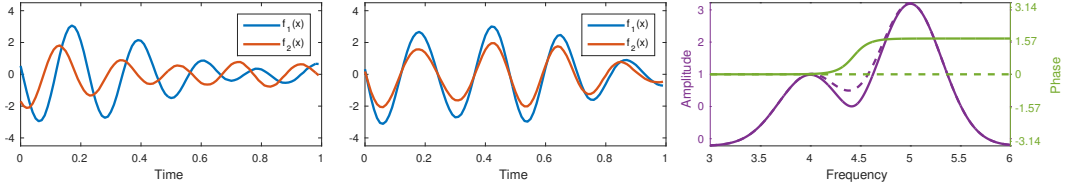

**Figure 1:** Latent functions drawn for two channels $f_1(\boldsymbol{x})$ (blue) and $f_2(\boldsymbol{x})$ (red) using the CSM kernel (**left**) and rank-1 SM-LMC kernel (**center**). The functions are comprised of two SG components centered at 4 and 5 Hz. For the CSM kernel, we set the phase shift $\psi_{c',2} = \pi$. **Right**: the cross-amplitude (purple) and cross-phase (green) spectra between $f_1(\boldsymbol{x})$ and $f_2(\boldsymbol{x})$ are shown for the CSM kernel (solid) and SM-LMC kernel (dashed). The ability to tune phase relationships is beneficial for kernel design and interpretation.

The moniker for the SM kernel in (6) reflects the mixture of Gaussian components that define the spectral density of the kernel. The SM kernel is able to represent any stationary covariance kernel given large enough $Q$; to name a few, this includes any combination of squared exponential, Matèrn, rational quadratic, or periodic kernels [9, 16, 24].

### 3.2 The Cross-Spectral Mixture Kernel

A multi-output version of the SM kernel uses the SG kernel directly within the LMC framework:

$$\boldsymbol{K}_{\text{SM-LMC}}(\tau; \boldsymbol{\theta}) = \sum_{q=1}^{Q} \boldsymbol{B}_q k_{\text{SG}}(\tau; \boldsymbol{\theta}_q), \tag{7}$$

where $Q$ SG kernels are shared among the outputs via the coregionalization matrices $\{\boldsymbol{B}_q\}_{q=1}^{Q}$. A generalized, non-stationary version of this SM-LMC kernel was proposed in [23] using the Gaussian process regression network (GPRN) [26]. The marginal distribution for any single channel is simply a Gaussian process with a SM covariance kernel. While this formulation is capable of providing a full cross-amplitude spectrum between two channels, it contains no information about a cross-phase spectrum. Specifically, each channel is merely a weighted sum of $\sum_q R_q$ latent functions where $R_q = \text{rank}(\boldsymbol{B}_q)$. Whereas these functions are shared exactly across channels, our novel CSM kernel shares phase-shifted versions of these latent functions across channels.

**Definition 3.1.** The cross-spectral mixture (CSM) kernel takes the form

$$k_{\text{CSM}}^{c,c'}(\tau; \boldsymbol{\theta}) = \sum_{q=1}^{Q} \sum_{r=1}^{R_q} \sqrt{a_{cq}^r a_{c'q}^r} \exp(-\frac{1}{2}\nu_q \tau^2) \cos\left(\mu_q\left(\tau + \phi_{c'q}^r - \phi_{cq}^r\right)\right), \tag{8}$$

where $\boldsymbol{\theta} = \{\nu_q, \mu_q, \{\boldsymbol{a}_q^r, \phi_q^r, \phi_{1q}^r \triangleq 0\}_{r=1}^{R_q}\}_{q=1}^{Q}$ has $2Q + \sum_{q=1}^{Q} R_q(2C-1)$ degrees of freedom, and $a_{cq}^r$ and $\phi_{cq}^r$ respectively represent the amplitude and shift in the input space for latent functions associated with channel $c$. In the LMC framework, the CSM kernel is

$$\boldsymbol{K}_{\text{CSM}}(\tau; \boldsymbol{\theta}) = \text{Re}\left\{\sum_{q=1}^{Q} \boldsymbol{B}_q \widetilde{k}_{\text{SG}}(\tau; \boldsymbol{\theta}_q)\right\}, \qquad \boldsymbol{B}_q = \sum_{r=1}^{R_q} \boldsymbol{\beta}_q^r (\boldsymbol{\beta}_q^r)^{\dagger},$$

$$\widetilde{k}_{\text{SG}}(\tau; \boldsymbol{\theta}_q) = \exp(-\frac{1}{2}\nu_q \tau^2 + j\mu_q \tau), \qquad \beta_{cq}^r = \sqrt{a_{cq}^r} \exp(-j\psi_{cq}^r),$$

where $\widetilde{k}_{\text{SG}}(\tau, \boldsymbol{\theta}_q)$ is phasor notation of the SG kernel, $\boldsymbol{B}_q$ is rank-$R_q$, $\{\beta_{cq}^r\}$ are complex scalar coefficients encoding amplitude and phase, and $\psi_{cq}^r \triangleq \mu_q \phi_{cq}^r$ is an alternative phase representation. We use complex notation where $j = \sqrt{-1}$, $\text{Re}\{\cdot\}$ returns the real component of its argument, and $\boldsymbol{\beta}^{\dagger}$ represents the complex conjugate of $\boldsymbol{\beta}$.

Both the CSM and SM-LMC kernels force the marginal distribution of data from a single channel to be a Gaussian process with a SM covariance kernel. The CSM kernel is derived in the Supplemental Material by considering functions represented by phase-shifted sinusoidal signals, $f_c(x) = \sum_{q=1}^{Q} \sum_{r=1}^{R_q} \sqrt{2a_{cq}^r} \cos(\omega_q^r(x + \phi_{cq}^r))$, where each $\omega_q^r \stackrel{iid}{\sim} \mathcal{N}(\mu_q, \nu_q)$. Computing the cross-covariance function $\text{cov}(f_c(x), f_{c'}(x + \tau))$ provides the CSM kernel.

A comparison between draws from Gaussian processes with CSM and SM-LMC kernels is shown in Figure 1. The utility of the CSM kernel is clearly illustrated by its ability to encode phase

information, as well as its powerful *functional form* of the full cross-spectrum (both amplitude and phase). The amplitude function $A_{c,c'}(\omega)$ and phase function $\Phi_{c,c'}(\omega)$ are obtained by representing the cross-spectrum in phasor notation, i.e., $\Gamma_{c,c'}(\omega; \boldsymbol{\Theta}) = \sum_q (\boldsymbol{B}_q)_{c,c'} S_{SG}(\omega; \boldsymbol{\theta}_q) = A_{c,c'}(\omega) \exp(j\Phi_{c,c'}(\omega))$. Interestingly, while the CSM and SM-LMC kernels have identical marginal amplitude spectra for shared $\{\mu_q, \nu_q, \boldsymbol{a}_q\}$, their cross-amplitude spectra differ due to the inherent destructive interference of the CSM kernel (see Figure 1, right).

## 4 Multi-Channel HMM Analysis

Neuroscientists are interested in examining how the network structure of the brain changes as animals undergo a task, or various levels of arousal [15]. The LFP signal is a modality that allows researchers to explore this network structure. In the model provided in this section, we cluster segments of the LFP signal into discrete "brain states" [21]. Each brain state is represented by a unique cross-spectrum provided by the CSM kernel. The use of the full cross-spectrum to define brain states is supported by previous work discovering that 1) the power spectral density of LFP signals indicate various levels of arousal states in mice [7, 21], and 2) frequency-dependent phase synchrony patterns change as animals undergo different conditions in a task [11, 17, 18] (see Figure 2).

The vector-valued observations from $C$ channels are segmented into $W$ contiguous, non-overlapping windows. The windows are common across channels, such that the $C$-channel data for window $w \in \{1, \ldots, W\}$ are represented by $\boldsymbol{y}_n^w = [y_{n1}^w, \ldots, y_{nC}^w]^T$ at sample location $x_n^w$. Given data, each window consists of $N_w$ temporal samples, but the model is defined for any set of sample locations.

We model the observations $\{\boldsymbol{y}_n^w\}$ as emissions from a hidden Markov model (HMM) with $L$ hidden, discrete states. State assignments are represented by latent variables $\zeta_w \in \{1, \ldots, L\}$ for each window $w \in \{1, \ldots, W\}$. In general, $L$ is a set upper bound of the number of states (brain states [21], or "clusters"), but the model can shrink down and infer the number of states needed to fit the data. This is achieved by defining the dynamics of the latent states according to a Bayesian HMM [14]:

$$\zeta_1 \sim \text{Categorical}(\boldsymbol{\rho}_0), \qquad \zeta_w \sim \text{Categorical}(\boldsymbol{\rho}_{\zeta_{w-1}}) \ \forall w \geq 2, \qquad \boldsymbol{\rho}_0, \boldsymbol{\rho}_\ell \sim \text{Dirichlet}(\boldsymbol{\nu}),$$

where the initial state assignment is drawn from a categorical distribution with probability vector $\boldsymbol{\rho}_0$ and all subsequent states assignments are drawn from the transition vector $\boldsymbol{\rho}_{\zeta_{w-1}}$. Here, $\rho_{\ell h}$ is the probability of transitioning from state $\ell$ to state $h$. The vectors $\{\boldsymbol{\rho}_0, \boldsymbol{\rho}_1, \ldots, \boldsymbol{\rho}_L\}$ are independently drawn from symmetric Dirichlet distributions centered around $\boldsymbol{\nu} = [1/L, \ldots, 1/L]$ to impose sparsity on transition probabilities. In effect, this allows the model to learn the number of states needed for the data (i.e., fewer than $L$) [3].

Each cluster $\ell \in \{1, \ldots, L\}$ is assigned GP parameters $\boldsymbol{\theta}_\ell$. The latent cluster assignment $\zeta_w$ for window $w$ indicates which set of GP parameters control the emission distribution of the HMM:

$$\boldsymbol{y}_n^w \sim \mathcal{N}(\boldsymbol{f}_w(x_n^w), \boldsymbol{H}_{\zeta_w}^{-1}), \qquad\qquad \boldsymbol{f}_w(x) \sim \mathcal{GP}(\boldsymbol{0}, \boldsymbol{K}(x, x'; \boldsymbol{\theta}_{\zeta_w})), \qquad (9)$$

where $(\boldsymbol{K}(x, x'; \boldsymbol{\theta}_\ell))_{c,c'} = k_{\text{CSM}}^{c,c'}(x, x'; \boldsymbol{\theta}_\ell)$ is the CSM kernel, and the cluster-dependent precision $\boldsymbol{H}_{\zeta_w} = \text{diag}(\boldsymbol{\eta}_{\zeta_w})$ generates independent Gaussian observation noise. In this way, each window $w$ is modeled as a stochastic process with a multi-channel cross-spectrum defined by $\boldsymbol{\theta}_{\zeta_w}$.

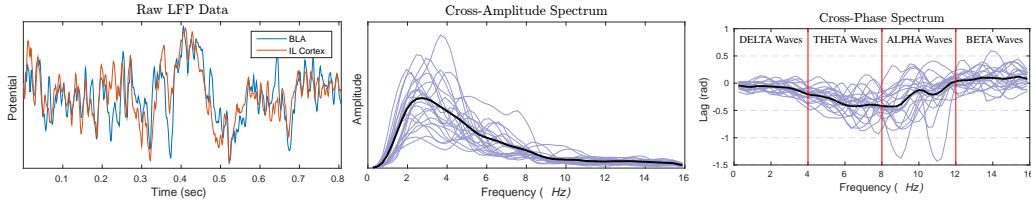

**Figure 2:** A short segment of LFP data recorded from the basolateral amygdala and infralimbic cortex is shown on the **left**. The cross-amplitude and phase spectra are produced using Welch's averaged periodogram method [22] for several consecutive 5 second windows of LFP data. Frequency dependent phase synchrony lags are consistently present in the cross-phase spectrum, motivating the CSM kernel. This frequency dependency aligns with preconceived notions of bands, or brain waves (e.g., 8-12 *Hz* alpha waves).

# 5 Inference

A convenient notation vectorizes all observations within a window, $\boldsymbol{y}^w = \text{vec}([\boldsymbol{y}_1^w, \ldots, \boldsymbol{y}_{N_w}^w]^T)$, where $\text{vec}(\boldsymbol{A})$ is the vectorization of matrix $\boldsymbol{A}$; i.e., the first $N_w$ elements of $\boldsymbol{y}^w$ are observations from channel 1, up to the last $N_w$ elements of $\boldsymbol{y}^w$ belonging to channel $C$. Because samples are obtained on an evenly spaced temporal grid, we fix $N_w = N$ and align relative sample locations within a window to an oracle $\boldsymbol{x}^w = \boldsymbol{x} = [x_1, \ldots, x_N]^T$ for all $w$.

The model in Section 4 generates the set of observations $\boldsymbol{Y} = \{\boldsymbol{y}^w\}_{w=1}^W$ at aligned sample locations $\boldsymbol{x}$ given kernel hyperparameters $\boldsymbol{\Theta} = \{\boldsymbol{\theta}_\ell, \boldsymbol{\eta}_\ell\}_{\ell=1}^L$ and model variables $\boldsymbol{\Omega} = \{\{\boldsymbol{\rho}_\ell\}_{\ell=0}^L, \{\zeta_w\}_{w=1}^W\}$. The latent variables $\boldsymbol{\Omega}$ are inverted using mean-field variational inference [3], obtaining an approximate posterior distribution $q(\boldsymbol{\Omega}) = q(\zeta_{1:W}) \prod_{\ell=0}^L \text{Dir}(\boldsymbol{\rho}_\ell; \boldsymbol{\alpha}_\ell)$. The approximate posterior is chosen to minimize the KL divergence to the true posterior distribution $p(\boldsymbol{\Omega}|\boldsymbol{Y}, \boldsymbol{\Theta}, \boldsymbol{x})$ using the standard variational EM method detailed in Chapter 3 of [3].

During each iteration of the variational EM algorithm, the kernel hyperparameters $\boldsymbol{\Theta}$ are chosen to maximize the expected marginal log-likelihood $\mathcal{Q} = \sum_{w=1}^W \sum_{\ell=1}^L q(\zeta_w = \ell) \log \mathcal{N}(\boldsymbol{y}^w; \boldsymbol{0}, \boldsymbol{\Gamma}_\ell)$ via gradient ascent, where $q(\zeta_w = \ell)$ is the marginal posterior probability that window $w$ is assigned to brain state $\ell$, and $\boldsymbol{\Gamma}_\ell = \text{Re}\{\widetilde{\boldsymbol{\Gamma}}_\ell\}$ is the CSM kernel matrix for state $\ell$ with the complex form $\widetilde{\boldsymbol{\Gamma}}_\ell = \sum_q \boldsymbol{B}_q^\ell \otimes \widetilde{k}_{\text{SG}}(\boldsymbol{x}, \boldsymbol{x}; \boldsymbol{\theta}_\ell) + \boldsymbol{H}_\ell^{-1} \otimes \boldsymbol{I}_N$. Performing gradient ascent requires the derivatives $\frac{\partial \mathcal{Q}}{\partial \boldsymbol{\Theta}_j} = \frac{1}{2} \sum_{w,\ell} \text{tr}((\boldsymbol{\alpha}_{\ell w} \boldsymbol{\alpha}_{\ell w}^T - \boldsymbol{\Gamma}_\ell^{-1}) \frac{\partial \boldsymbol{\Gamma}_\ell}{\partial \boldsymbol{\Theta}_j})$ where $\boldsymbol{\alpha}_{\ell w} = \boldsymbol{\Gamma}_\ell^{-1} \boldsymbol{y}^w$ [16]. A naïve implementation of this gradient requires the inversion of $\boldsymbol{\Gamma}_\ell$, which has complexity $\mathcal{O}(N^3 C^3)$ and storage requirements $\mathcal{O}(N^2 C^2)$ since a simple method to invert a sum of Kronecker products does not exist.

A common trick for GPs with evenly spaced samples (e.g., a temporal grid) is to use the discrete Fourier transform (DFT) to approximate the inverse of $\boldsymbol{\Gamma}_\ell$ by viewing this as an approximately circulant matrix [5, 12]. These methods can speed up inference because circulant matrices are diagonalizable by the DFT coefficient matrix. Adjusting these methods to the multi-output formulation, we show how the DFT of the marginal covariance matrices retains the cross-spectrum information.

**Proposition 5.1.** Let $\boldsymbol{y}^w \sim \mathcal{N}(\boldsymbol{0}, \boldsymbol{\Gamma}_{\zeta_w})$ represent the marginal likelihood of circularly-symmetric [8] real-valued observations in window $w$, and denote the concatenation of the DFT of each channel as $\boldsymbol{z}^w = (\boldsymbol{I}_C \otimes \boldsymbol{U})^\dagger \boldsymbol{y}^w$ where $\boldsymbol{U}$ is the $N \times N$ unitary DFT matrix. Then, $\boldsymbol{z}^w$ is shown in the Supplemental Material to have the complex normal distribution [8]:

$$\boldsymbol{z}^w \sim \mathcal{CN}(\boldsymbol{0}, 2\boldsymbol{S}_{\zeta_w}), \qquad \boldsymbol{S}_\ell = \delta^{-1} \sum_{q=1}^Q \boldsymbol{B}_q^\ell \otimes \boldsymbol{W}_q^\ell + \boldsymbol{H}_\ell^{-1} \otimes \boldsymbol{I}_N, \qquad (10)$$

where $\delta = x_{i+1} - x_i$ for all $i = 2, \ldots, N$, and $\boldsymbol{W}_q^\ell \approx \text{diag}([S_{\text{SG}}(\boldsymbol{\omega}; \boldsymbol{\theta}_{\ell q}), \boldsymbol{0}])$ is approximately diagonal. The spectral density $S_{\text{SG}}(\boldsymbol{\omega}; \boldsymbol{\theta}) = [S_{\text{SG}}(\omega_1; \boldsymbol{\theta}), \ldots, S_{\text{SG}}(\omega_{\lfloor \frac{N+1}{2} \rfloor}; \boldsymbol{\theta})]$ is found via (4) at angular frequencies $\boldsymbol{\omega} = \frac{2\pi}{N\delta} [0, 1, \ldots, \lfloor \frac{N}{2} \rfloor]$, and $\boldsymbol{0} = [0, \ldots, 0]$ is a row vector of $\lfloor \frac{N-1}{2} \rfloor$ zeros.

The hyperparameters of the CSM kernels $\boldsymbol{\Theta}$ may now be optimized from the expected marginal log-likelihood of $\boldsymbol{Z} = \{\boldsymbol{z}^w\}_{w=1}^W$ instead of $\boldsymbol{Y}$. Conceptually, the only difference during the fitting process is that, with the latter, derivatives of the covariance kernel are used, while, with the former, derivatives of the power spectral density are used. Computationally, this method improves the naïve $\mathcal{O}(N^3 C^3)$ complexity of fitting the standard CSM kernel to $\mathcal{O}(N C^3)$ complexity. Memory requirements are also reduced from $\mathcal{O}(N^2 C^2)$ to $\mathcal{O}(N C^2)$. The reason for this improvement is that $\boldsymbol{S}_\ell$ is now represented as $N$ independent $C \times C$ blocks, reducing the inversion of $\boldsymbol{S}_\ell$ to inverting a permuted block-diagonal matrix.

# 6 Experiments

Section 6.1 demonstrates the performance of the CSM kernel and the accuracy of the DFT approximation In Section 6.2, the DFT approximation for the CSM kernel is used in a Bayesian HMM framework to cluster time-varying multi-channel LFP data based on the full cross-spectrum; the HMM states here correspond to states of the brain during LFP recording.

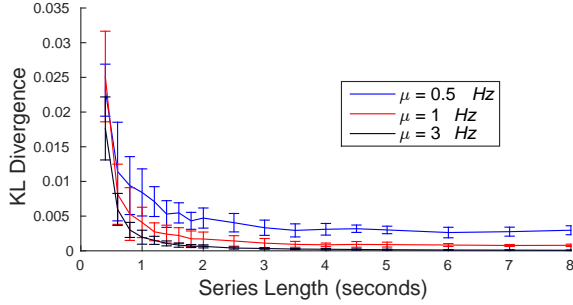

**Figure 3:** Time-series data is drawn from a Gaussian process with a known CSM covariance kernel, where the domain restricted to a fixed number of seconds. A Gaussian process is then fitted to this data using the DFT approximation. The KL-divergence of the fitted marginal likelihood from the true marginal likelihood is shown.

**Table 1:** The mean and standard deviation of the difference between the AIC value of a given model and the AIC value of the rank-2 CSM model. Lower values are better.

| Rank | Model | $\Delta$ AIC |
|------|-------|--------------|
| 1 | SE-LMC | 4770 (993) |
| 1 | SM-LMC | 512 (190) |
| 1 | CSM | 109 (110) |
| 2 | SE-LMC | 5180 (1120) |
| 2 | SM-LMC | 325 (167) |
| 2 | CSM | **0 (0)** |
| 3 | SE-LMC | 5550 (1240) |
| 3 | SM-LMC | 412 (184) |
| 3 | CSM | 204 (71.7) |

## 6.1 Performance and Inference Analysis

The performance of the CSM kernel is compared to the SM-LMC kernel and SE-LMC (squared exponential) kernel. Each of these models allow Q=20, and the rank of the coregionalization matrices is varied from rank-1 to rank-3. For a given rank, the CSM kernel always obtains the largest marginal likelihood for a window of LFP data, and the marginal likelihood always increases for increasing rank. To penalize the number of kernel parameters (e.g., a rank-3, Q=20 CSM kernel for 7 channels has 827 free parameters to optimize), the Akaike information criterion (AIC) is used for model selection [1]. For this reason, we do not test rank greater than 3. Table 1 shows that a rank-2 CSM kernel is selected using this criterion, followed by a rank-1 CSM kernel. To show the rank-2 CSM kernel is consistently selected as the preferred model we report means and standard deviations of AIC value differences across 30 different randomly selected 3-second windows of LFP data.

Next, we provide numerical results for the conditions required when using the DFT approximation in (10). This allows for one to define details of a particular application in order to determine if the DFT approximation to the CSM kernel is appropriate. A CSM kernel is defined for two outputs with a single Gaussian component, $Q = 1$. The mean frequency and variance for this component are set to push the limits of the application. For example, with LFP data, low frequency content is of interest, namely greater than 1 $Hz$; therefore, we test values of $\widetilde{\mu}_1 \in \{\frac{1}{2}, 1, 3\}$ $Hz$. We anticipate variances at these frequencies to be around $\widetilde{\nu}_1 = 1$ $Hz^2$. A conversion to angular frequency gives $\mu_1 = 2\pi\widetilde{\mu}_1$ and $\nu_1 = 4\pi^2\widetilde{\nu}_1$. The covariance matrix $\boldsymbol{\Gamma}$ in (3) is formed using these parameters, a fixed noise variance, and $N$ observations on a time grid with sampling rate of 200 $Hz$. Data $\boldsymbol{y}$ are drawn from the marginal likelihood with covariance $\boldsymbol{\Gamma}$.

A new CSM kernel is fit to $\boldsymbol{y}$ using the DFT approximation, providing an estimate $\hat{\boldsymbol{\Gamma}}$. The KL divergence of the fitted marginal likelihood from the true marginal likelihood is

$$\text{KL}(p(\boldsymbol{y}|\hat{\boldsymbol{\Gamma}})||p(\boldsymbol{y}|\boldsymbol{\Gamma})) = \frac{1}{2}\left[\log\frac{|\boldsymbol{\Gamma}|}{|\hat{\boldsymbol{\Gamma}}|} - N + \text{tr}(\boldsymbol{\Gamma}^{-1}\hat{\boldsymbol{\Gamma}})\right],$$

where $|\cdot|$ and $\text{tr}(\cdot)$ are the determinant and trace operators, respectively. Computing $\frac{1}{N}\text{KL}(p(\boldsymbol{y}|\hat{\boldsymbol{\Gamma}})||p(\boldsymbol{y}|\boldsymbol{\Gamma}))$ for various values of $\widetilde{\mu}_1$ and $N$ provides the results in Figure 3. This plot shows that the DFT approximation struggles to resolve low frequency components unless the series length is sufficiently long. Due to the approximation error, when using the DFT approximation on LFP data we *a priori* filter out frequencies below 1.5 $Hz$ and perform analyses with a series length of 3 seconds. This ensures the DFT approximation represents the true covariance matrix. The following application of the CSM kernel uses these settings.

## 6.2 Including the CSM Kernel in a Bayesian Hierarchical Model

We analyze 12 hours of LFP data of a mouse transitioning between different stages of sleep [7, 21]. Observations were recorded simultaneously from 4 channels [6], high-pass filtered at 1.5 $Hz$, and subsampled to 200 $Hz$. Using 3 second windows provides $N = 600$ and $W = 14,400$. The HMM was implemented with the number of kernel components $Q = 15$ and the number of states $L = 7$.

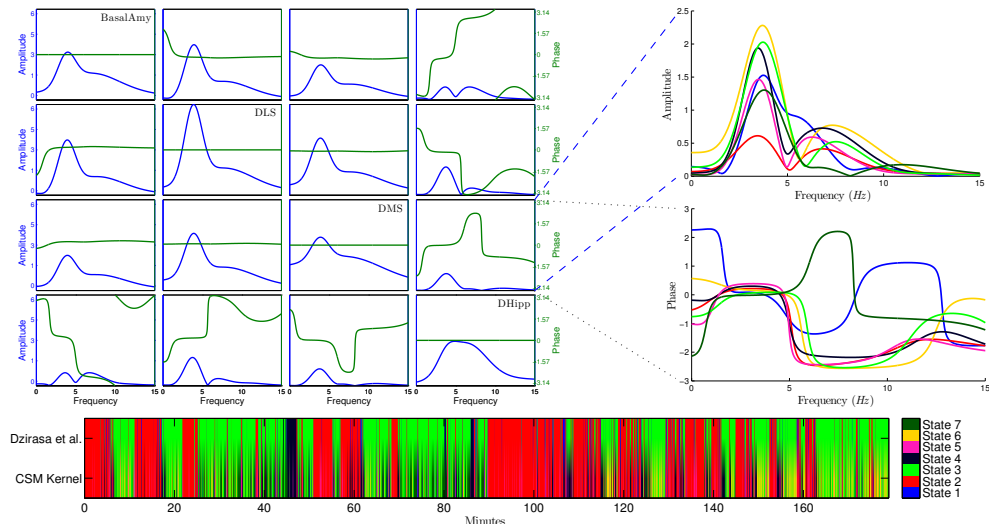

**Figure 4:** A subset of results from the Bayesian HMM analysis of brain states. In the **upper left**, the full cross-spectrum for an arbitrary state (state 7) is plotted. In the **upper right**, the amplitude (**top**) and phase (**bottom**) functions for the cross-spectrum between the Dorsomedial Striatum (DMS) and Hippocampus (DHipp) are shown for *all* seven states. On the **bottom**, the maximum likelihood state assignments are shown and compared to the state assignments from [7]. The same colors between the CSM state assignments and the phase and amplitude functions correspond to the same state. These colors are alligned to the [7] states, but there is no explicit relationship between the colors of the two state sequences.

This was chosen because sleep staging tasks categorize as many as seven states: various levels of rapid eye movement, slow wave sleep, and wake [20]. Although rigorous model selection on $L$ is necessary to draw scientific conclusions from the results, the purpose of this experiment is to illustrate the utility of the CSM kernel in this application.

An illustrative subset of the results are shown in Figure 4. The full cross-spectrum is shown for a single state (state 7), and the cross-spectrum between the Dorsomedial Striatum and the Dorsal Hippocampus are shown for all states. Furthermore, we show the progression of these brain state assignments over 3 hours and compare them to states from the method of [7], where statistics of the Hippocampus spectral density were clustered in an *ad hoc* fashion. To the best of our knowledge, this method represents the most relevant and accurate results for sleep staging from LFP signals in the neuroscience literature. From these results, it is apparent that our clusters pick up sub-states of [7]. For instance, states 3, 6, and 7 all appear with high probability when the method from [7] infers state 3. Observing the cross-phase function of sub-state 7 reveals striking differences from other states in the theta wave (4-7 *Hz*) and the alpha wave (8-15 *Hz*). This cross-phase function is nearly identical for states 2 and 5, implying that significant differences in the cross-amplitude spectrum may have played a role in identifying the difference between these two brain states.

Many more of these interesting details exist due to the expressive nature of the CSM kernel. As a full interpretation of the cross-spectrum results is not the focus of this work, we contend that the CSM kernel has the potential to have a tremendous impact in fields such as neuroscience, where the dynamics of cross-spectrum relationships of LFP signals are of great interest.

## 7   Conclusion

This work introduces the cross-spectral mixture kernel as an expressive kernel capable of extracting patterns for multi-channel observations. Combined with the powerful nonparametric representation of a Gaussian process, the CSM kernel expresses a *functional form* for every pairwise cross-spectrum between channels. This is a novel approach that merges Gaussian processes in the machine learning community to standard signal processing techniques. We believe the CSM kernel has the potential to impact a broad array of disciplines since the kernel can trivially be extended to any time-series application where Gaussian processes and the cross-spectrum are of interest.

**Acknowledgments**

The research reported here was funded in part by ARO, DARPA, DOE, NGA and ONR.

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
