[Supplementary Material · NIPS2015_CSMkernel_SM.pdf]

# GP Kernels for Cross-Spectrum Analysis
# Supplemental Material

[1]**Kyle Ulrich**, [3]**David E. Carlson**, [2]**Kafui Dzirasa**, [1]**Lawrence Carin**
[1]Department of Electrical and Computer Engineering, Duke University
[2]Department of Psychiatry and Behavioral Sciences, Duke University
[3]Department of Statistics, Columbia University
{kyle.ulrich, kafui.dzirasa, lcarin}@duke.edu
david.edwin.carlson@gmail.com

## A    Kernel Derivations

In this section, we detail derivations of the SG, SM, and CSM kernels by the method of the auto-covariance of periodic functions with unknown frequency.

### A.1    The Spectral Gaussian (SG) Kernel

Consider a periodic signal $f(x) = \sqrt{2}\cos(\omega(x + \phi))$ with the prior $\omega \sim \mathcal{N}(\mu, \nu)$ on the angular frequency. Letting $\boldsymbol{\theta} = \{\mu, \nu\}$, the stationary, positive definite auto-covariance function for $f(x)$ is derived according to

$$
\begin{aligned}
k_{\text{SG}}(\tau; \boldsymbol{\theta}) &\triangleq \text{cov}(f(x), f(x + \tau)) \\
&= \mathbb{E}\left[2\cos(\omega(x + \phi))\cos(\omega(x + \phi + \tau))\right] \\
&= \mathbb{E}\left[\cos(\omega\tau) - \cos(\omega(2x + 2\phi + \tau))\right] \\
&= \mathbb{E}_{p(\omega)}\left[\cos(\omega\tau)\right] \\
&= \int_{-\infty}^{\infty} \cos(\omega\tau)\frac{1}{\sqrt{2\pi\nu}}\exp\left(-\frac{1}{2\nu}(\omega - \mu)^2\right)d\omega \qquad (1) \\
&= \frac{1}{\sqrt{2\pi\nu}}\text{Re}\left\{\int_{-\infty}^{\infty}\exp\left(j\omega\tau - \frac{1}{2\nu}(\omega - \mu)^2\right)d\omega\right\}
\end{aligned}
$$

Let $a = \dfrac{1}{2\nu}, b = -\dfrac{1}{2}\left(j\tau + \dfrac{\mu}{\nu}\right)$, and $c = -\dfrac{\nu^2}{2\nu}$, then

$$
\begin{aligned}
&= \frac{1}{\sqrt{2\pi\nu}}\text{Re}\left\{\int_{-\infty}^{\infty}\exp(-a\omega^2 - 2b\omega + c)d\omega\right\} \\
&= \frac{1}{\sqrt{2\pi\nu}}\text{Re}\left\{\sqrt{\frac{\pi}{a}}\exp\left(\frac{b^2}{a} + c\right)\right\} \\
&= \text{Re}\left\{\exp(j\mu\tau - \frac{1}{2}\nu\tau^2)\right\} \\
&= \exp(-\frac{1}{2}\nu\tau^2)\cos(\mu\tau). \qquad (2)
\end{aligned}
$$

This results in an interpretable auto-covariance function, where $\mu$ represents the primary frequency and the variance $\nu$ controls how quickly this frequency dependency decays with $\tau$. Used as a stationary kernel in a Gaussian process, the SG kernel discovers a single Gaussian component in the spectral density.

## A.2 The Spectral Mixture (SM) Kernel

Consider a periodic signal $f(\boldsymbol{x}) = \sum_{q=1}^{Q} \sqrt{2a_q}\cos(\boldsymbol{\omega}_q^T(\boldsymbol{x} + \boldsymbol{\phi}_q))$, where $Q$ latent functions are superimposed, each with unique input-space offset $\boldsymbol{\phi}_q \in \mathbb{R}^P$ and angular frequency vector $\boldsymbol{\omega}_q \in \mathbb{R}^P$. Each angular frequency component is assigned the prior $\omega_q^{(p)} \sim \mathcal{N}(\mu_q^{(p)}, \nu_q^{(p)})$. Letting $\boldsymbol{\theta} = \{\boldsymbol{\mu}_q, \boldsymbol{\nu}_q\}_{q=1}^{Q}$ and $p(\boldsymbol{\Omega}) = \prod_q \prod_p p(\omega_q^{(p)})$, the auto-covariance function for $f(\boldsymbol{x})$ is derived according to

$$
\begin{aligned}
k_{\text{SM}}(\boldsymbol{\tau}; \boldsymbol{\theta}) &= \text{cov}(f(\boldsymbol{x}), f(\boldsymbol{x} + \boldsymbol{\tau})) \\
&= \mathbb{E}\left[2\sum_{q=1}^{Q}\sum_{r=1}^{Q}\sqrt{a_q a_r}\cos(\boldsymbol{\omega}_q^T(\boldsymbol{x} + \boldsymbol{\phi}_q))\cos(\boldsymbol{\omega}_r^T(\boldsymbol{x} + \boldsymbol{\phi}_r + \boldsymbol{\tau}))\right] \\
&= \sum_{q=1}^{Q} a_q \mathbb{E}_{p(\boldsymbol{\Omega})}\left[\cos(\boldsymbol{\omega}_q^T\boldsymbol{\tau})\right],
\end{aligned}
$$

where the expectation over the input domain eliminates cross terms between all $q \neq r$ since each $\omega_q^{(p)} \neq \omega_r^{(p)}$ with probability 1. Proceeding in the same fashion as the SG kernel,

$$
\begin{aligned}
k_{\text{SM}}(\boldsymbol{\tau}; \boldsymbol{\theta}) &= \sum_{q=1}^{Q} a_q \int_{-\infty}^{\infty}\cos(\boldsymbol{\omega}^T\boldsymbol{\tau})\prod_{p=1}^{P}\frac{1}{\sqrt{2\pi\nu_q^{(p)}}}\exp\left(-\frac{1}{2\nu_q^{(p)}}(\omega_q^{(p)} - \mu_q^{(p)})^2\right)d\boldsymbol{\omega} \\
&= \sum_{q=1}^{Q} a_q \prod_{p=1}^{P}\frac{1}{\sqrt{2\pi\nu_q^{(p)}}}\text{Re}\left\{\int_{-\infty}^{\infty}\exp\left(j\omega_q^{(p)}\tau_p - \frac{1}{2\nu_q^{(p)}}(\omega_q^{(p)} - \mu_q^{(p)})^2\right)d\omega_q^{(p)}\right\} \\
&= \sum_{q=1}^{Q} a_q \prod_{p=1}^{P}\exp(-\tfrac{1}{2}\nu_q^{(p)}\tau_p^2)\cos(\mu_q^{(p)}\tau_p)
\end{aligned}
\tag{3}
$$

where the final step proceeds by recognizing that each integral in the product is an SG component from (1). This is known as the spectral mixture (SM) kernel.

## A.3 The Cross-Spectral Mixture (CSM) Kernel

We now consider $C$ different channels and allow observations from each channel $\{f_c(\boldsymbol{x})\}_{c=1}^{C}$ to be represented as a linear combination of latent signals, $f_c(\boldsymbol{x}) = \sum_{q=1}^{Q}\sqrt{2a_{cq}}\cos(\boldsymbol{\omega}_q^T(\boldsymbol{x} + \boldsymbol{\phi}_{cq}))$. While the angular frequency components are still assigned the prior $\omega_q^{(p)} \sim \mathcal{N}(\mu_q^{(p)}, \nu_q^{(p)})$, each channel is assigned channel-specific input-space shifts $\boldsymbol{\phi}_{cq}$. When computing the cross-covariance function, these shifts will not cancel when $c_1 \neq c_2$. The cross-covariance function is derived according to

$$
\begin{aligned}
k_{\text{CSM}}^{c_1, c_2}(\boldsymbol{\tau}; \boldsymbol{\theta}) &= \text{cov}(f_{c_1}(\boldsymbol{x}), f_{c_2}(\boldsymbol{x} + \boldsymbol{\tau})) \\
&= \mathbb{E}\left[2\sum_{q=1}^{Q}\sum_{r=1}^{Q}\sqrt{a_{c_1 q}a_{c_2 r}}\cos(\boldsymbol{\omega}_q^T(\boldsymbol{x} + \boldsymbol{\phi}_{c_1 q}))\cos(\boldsymbol{\omega}_r^T(\boldsymbol{x} + \boldsymbol{\phi}_{c_2 r} + \boldsymbol{\tau}))\right] \\
&= \sum_{q=1}^{Q}\sqrt{a_{c_1 q}a_{c_2 q}}\,\mathbb{E}_{p(\boldsymbol{\Omega})}\left[\cos(\boldsymbol{\omega}_q^T\boldsymbol{\tau} + \boldsymbol{\phi}_{c_2 q} - \boldsymbol{\phi}_{c_1 q})\right] \\
&= \sum_{q=1}^{Q}\sqrt{a_{c_1 q}a_{c_2 q}}\prod_{p=1}^{P}\frac{1}{\sqrt{2\pi\nu_q^{(p)}}}\text{Re}\left\{\int_{-\infty}^{\infty}\exp\left(j\omega_q^{(p)}\tau_p + j(\phi_{c_2 q}^{(p)}\right.\right. \\
&\qquad\qquad\qquad\qquad\left.\left. -\phi_{c_1 q}^{(p)}) - \frac{1}{2\nu_q^{(p)}}(\omega_q^{(p)} - \mu_q^{(p)})^2\right)d\omega_q^{(p)}\right\} \\
&= \sum_{q=1}^{Q}\sqrt{a_{c_1 q}a_{c_2 q}}\prod_{p=1}^{P}\exp\left(-\tfrac{1}{2}\nu_q^{(p)}\tau_p^2\right)\cos\left(\mu_q^{(p)}\left(\tau_p + \phi_{c_2 q}^{(p)} - \phi_{c_1 q}^{(p)}\right)\right),
\end{aligned}
\tag{4}
$$

where the final step proceeds by recognizing that each integral in the product is simply an SG component from (1). The phase difference term $\phi_{c_2q}^{(p)} - \phi_{c_1q}^{(p)}$ in this last step simply gets added to the constant $c$ to produce the final result. We have termed this novel kernel the cross-spectral mixture (CSM) kernel.

## B  DFT Approximation for the CSM Kernel

Here, we detail the derivation of Proposition 5.1 in the main paper.

### B.1  Circulant Matrix Approximation

Dropping cluster assignments and window indicators, the marginal likelihood variance in the paper was defined as $\boldsymbol{\Gamma} = \text{Re}\{\widetilde{\boldsymbol{\Gamma}}\}$ where $\widetilde{\boldsymbol{\Gamma}} = \sum_q \boldsymbol{B}_q \otimes \boldsymbol{K}_q + \boldsymbol{H}^{-1} \otimes \boldsymbol{I}_N$ and $\boldsymbol{K}_q = \widetilde{k}_{\text{SG}}(\boldsymbol{x}, \boldsymbol{x}; \boldsymbol{\theta})$ for a set of input locations $\boldsymbol{x}$.

By definition, the covariance matrix $\boldsymbol{\Sigma} = \boldsymbol{K}_q$, for any $q$, is a symmetric Toeplitz matrix. There are several properties of symmetric Toeplitz matrices that make the form of $\widetilde{\boldsymbol{\Gamma}}$ particularly convenient. One of these properties is that a matrix of this form is uniquely identified by the first column of the matrix, denoted here as $\boldsymbol{\sigma}$. Furthermore, a symmetric Toeplitz matrix is closely related to a circulant matrix. Specifically, reflecting the first $\lfloor \frac{N}{2} + 1 \rfloor$ elements of $\boldsymbol{c}$ to the last $\lceil \frac{N}{2} + 1 \rceil$ elements will produce a circulant approximation, $\widetilde{\boldsymbol{\sigma}}$, to $\boldsymbol{\sigma}$. The resulting matrix is also Toeplitz, $\widetilde{\boldsymbol{\Sigma}} = \text{toeplitz}(\widetilde{\boldsymbol{\sigma}})$, and is closely related to the discrete Fourier transform (DFT) in the following way. Given the eigen-decomposition $\widetilde{\boldsymbol{\Sigma}} = \boldsymbol{V}_{\widetilde{\boldsymbol{\Sigma}}} \boldsymbol{\Lambda}_{\widetilde{\boldsymbol{\Sigma}}} \boldsymbol{V}_{\widetilde{\boldsymbol{\Sigma}}}^{-1}$, the vector of eigenvalues $\boldsymbol{\lambda}_{\widetilde{\boldsymbol{\Sigma}}} = \text{diag}(\boldsymbol{\Lambda}_{\widetilde{\boldsymbol{\Sigma}}})$ equals the complex DFT of $\widetilde{\boldsymbol{\sigma}}$. Mathematically, $\boldsymbol{U}\widetilde{\boldsymbol{\sigma}} = \boldsymbol{\lambda}_{\widetilde{\boldsymbol{\Sigma}}}$, where $\boldsymbol{U}$ is the $N \times N$ unitary DFT matrix [?].

This reveals that the columns of $\boldsymbol{U}$ are the eigenvectors of $\widetilde{\boldsymbol{\Sigma}}$, thereby implying that (a) $\widetilde{\boldsymbol{\Sigma}}$ is diagonalizable by the DFT coefficient matrix, and (b) the elements along the diagonal, $\boldsymbol{\lambda}_{\widetilde{\boldsymbol{\Sigma}}}$, refer to the power spectral density of $\widetilde{\boldsymbol{\sigma}}$, i.e.,

$$\boldsymbol{U}^{\dagger}\widetilde{\boldsymbol{\Sigma}}\boldsymbol{U} = \boldsymbol{\Lambda}_{\widetilde{\boldsymbol{\Sigma}}} = \text{diag}\left(\delta^{-1}S(\boldsymbol{\omega})\right), \tag{5}$$

where the spectral density values are scaled by $\delta^{-1}$ and evaluated independently $S(\boldsymbol{\omega}) = [S(\omega_1), \ldots, S(\omega_N)]$ at angular frequency locations $\boldsymbol{\omega} = \frac{2\pi}{N\delta}\left[0, 1, \ldots, \lfloor\frac{N}{2}\rfloor, -\lfloor\frac{N-1}{2}\rfloor, \ldots, -1\right]$. Because of this, the full matrix $\boldsymbol{\Sigma}$ need not ever be stored; rather, the spectral density vector $S(\boldsymbol{\omega})$ is sufficient.

As $\delta \to 0$ and $x_N \to \infty$, this spectral density is exact and there is no error in approximating $\boldsymbol{\Sigma}$ with $\widetilde{\boldsymbol{\Sigma}}$. This is due to the increasing resolution of the DFT frequency bins. When the resolution is poor, however, the elements of $\widetilde{\boldsymbol{\Sigma}}$ have small, but non-negligible, negative correlations.

### B.2  Marginal Likelihood of DFT Coefficients

The marginal likelihood formulation $\boldsymbol{y} \sim \mathcal{N}(\boldsymbol{0}, \boldsymbol{\Gamma})$ is particularly useful if the DFT approximation from the previous section is applied. Specifically, the marginal likelihood of the linear transformations $\{\boldsymbol{U}^{\dagger}\boldsymbol{y}_c\}_{c=1}^C$ is of interest. The following provides details about this distribution.

Although only the real portion of $\boldsymbol{y} = \boldsymbol{y}^r + j\boldsymbol{y}^i$ is observed, the fact that $\boldsymbol{y}^r$ is a sum of latent sinusoidal signals allows the complex vector $\boldsymbol{y}$ to be represented as a circularly symmetric complex normal random variable, such that $\boldsymbol{y} \sim \mathcal{CN}(\boldsymbol{0}, 2\widetilde{\boldsymbol{\Gamma}})$. This formulation implies that the distribution of $\boldsymbol{y}$ is invariant to any real phase shift in the underlying latent signals [?]. To avoid complex numbers, the joint distribution of the real and imaginary components of $\boldsymbol{y}$ is equivalent to the multivariate Gaussian distribution,

$$\begin{bmatrix} \boldsymbol{y}^r \\ \boldsymbol{y}^i \end{bmatrix} \sim \mathcal{N}\left(\begin{bmatrix} \boldsymbol{0} \\ \boldsymbol{0} \end{bmatrix}, \begin{bmatrix} \text{Re}\{\widetilde{\boldsymbol{\Gamma}}\} & \text{Im}\{-\widetilde{\boldsymbol{\Gamma}}\} \\ \text{Im}\{\widetilde{\boldsymbol{\Gamma}}\} & \text{Re}\{\widetilde{\boldsymbol{\Gamma}}\} \end{bmatrix}\right), \tag{6}$$

where the marginal distribution for the real component is $\boldsymbol{y}^r \sim \mathcal{N}(\boldsymbol{0}, \text{Re}\{\widetilde{\boldsymbol{\Gamma}}\})$, the exact form given above.

Denoting the DFT of each channel as $\boldsymbol{z} = (\boldsymbol{I}_C \otimes \boldsymbol{U})^\dagger \boldsymbol{y}$, and using the circulant approximation for each matrix $\{\boldsymbol{K}_q\}_{q=1}^Q$ along with (5) provides the equivalent marginal distribution for the DFT coefficients of $\boldsymbol{z} \sim \mathcal{CN}(\boldsymbol{0}, 2\boldsymbol{S})$, where

$$\boldsymbol{S} = (\boldsymbol{I}_C \otimes \boldsymbol{U})^\dagger \boldsymbol{\Gamma} (\boldsymbol{I}_C \otimes \boldsymbol{U})$$

$$\approx \boldsymbol{H}^{-1} \otimes \boldsymbol{I}_N + \delta^{-1} \sum_{q=1}^Q (\boldsymbol{\beta}_q \boldsymbol{\beta}_q^\dagger) \otimes \boldsymbol{\Lambda}_{\widetilde{\boldsymbol{K}}_q}. \tag{7}$$

To obtain this result, the property of Kronecker products $(\boldsymbol{A}_1 \otimes \boldsymbol{B}_1)(\boldsymbol{A}_2 \otimes \boldsymbol{B}_2) = (\boldsymbol{A}_1 \boldsymbol{A}_2) \otimes (\boldsymbol{B}_1 \boldsymbol{B}_2)$ was used, leaving the additive Gaussian noise term $\boldsymbol{H}^{-1} \otimes \boldsymbol{I}_N$ untouched (since $\boldsymbol{U}^\dagger \boldsymbol{U} = \boldsymbol{I}_N$) and replacing each $\boldsymbol{K}_q$ with its spectral density, $\delta^{-1} \boldsymbol{\Lambda}_{\widetilde{\boldsymbol{K}}_q}$.

One important comment pertains to the power spectral density along the diagonal of $\boldsymbol{\lambda}_{\widetilde{\boldsymbol{K}}_q}$. While the Fourier transform of any real data $\boldsymbol{y}^r$ results in a symmetric spectral density, the Fourier transform of circularly symmetric data $\boldsymbol{y}$ results in a spectral density that equals zero for half of the spectrum. Therefore, elements along the diagonal of $\boldsymbol{\Lambda}_{\widetilde{\boldsymbol{K}}_q}$ will equal zero for half the spectrum, i.e., the spectral density will contain only white noise for $\boldsymbol{\omega} = \frac{2\pi}{N\delta} \left[ -\lfloor \frac{N-1}{2} \rfloor, \dots, -1 \right]$.