[Reviews · NeurIPS 2015]

Submitted by Assigned_Reviewer_1

The authors propose a flexible and interpretable kernel (the CSM kernel), building on spectral mixture kernels, for learning relationships between multiple tasks.

The starting point is to use Gaussian processes with 1 component spectral mixture kernels as the basis functions in a linear model of coregionalisation (SM-LMC).

However, SM-LMC does not contain information about the phases between channels.

Thus the authors propose the cross spectral mixture kernel, which mixes phase shifted versions of spectral mixture kernels across channels.

The resulting kernel is interpretable and flexible.

The authors note that if they have an evenly spaced temporal grid, this gives rise to covariance matrix structure which can be exploited for fast computations.

In particular, they approximate their covariance matrices with circulant matrices, and use discrete Fourier transforms to speed up inference and learning.

They implement the CSM kernel as part of a Bayesian HMM, and apply the model to a neuroscience application.

The paper proposes a useful technically sound method, with a compelling application.

Detailed comments:

- A generalized version of the "SM-LMC" was discussed in chapter 4.4, p. 137 of Wilson [2014], "Covariance kernels for fast automatic pattern discovery and extrapolation with Gaussian processes", PhD thesis.

In particular, it was proposed to model $\bm{y}(x) = W(x) \bm{f}(x)$, where $\bm{y}(x)$ are the responses, $\bm{f}(x)$ is a vector of GPs each with 1 component SM kernels, and W(x) is an input dependent matrix of mixing weights.

Setting W(x) = W, one recovers the SM-LMC.

Quality:

The paper is technically sound, and the application is very compelling.

I would expect, however, the computational complexity to be O(C^3 N log N) rather than O(C^3 N).

An alternative to the proposed approach would be to exploit the Toeplitz structure of your matrices and perform linear conjugate gradients, which could be exact to within numerical precision, but somewhat slower than your approach.

Evaluating the log determinant would require O(N^2) computations and O(N) memory. By following an inference strategy similar to "Gaussian Process Regression Networks", ICML, you may be able to further reduce your complexity to O(C N).

In general, I was a little unsure of how much the scalability of the proposed approach was necessary for this application.

It would help to clarify this in a final version.

Clarity

The paper was a pleasure to read.

Originality

There is valuable novelty in decoupling the phase shifts between different tasks, as proposed in the CSM kernel, which usefully builds on the SM kernel.

Significance

I can easily imagine others finding this methodology widely applicable.

I really enjoyed the neuroscience application, and only wish the paper went into this application more deeply, with more interpretation of results.

It's very exciting that we can use these kernel learning approaches to gain new significant insights into our data, in addition to improved predictive performance.
Summary: A high quality paper, which proposes a scalable kernel learning strategy, building on spectral mixture kernels, for extracting interpretable relationships between multiple tasks, with a compelling neuroscience application.

Submitted by Assigned_Reviewer_2

The paper presents an extension of the spectral mixture kernel (Wilson and Adams, 2013) for multi-variate or multi-output Gaussian processes. The authors refer to this kernel as the cross-spectral mixture kernel. Since the kernel proposed is stationary, they also combine the GP model with hidden Markov models to make it quasi-stationary. They use the composite model for describing local field potentials.

Quality

I think the authors should decide better how do they want to "sell" their paper. Sometimes the paper looks like an extension of the spectral mixture kernel for multiple-outputs, and sometimes it looks like a very nice crafted model for a particular type of signals in neuroscience. I think the emphasis in the experimentation should reflect what do you want to show. It is a bit difficult when the paper attempts to do both, because there is not clear emphasis on the type of experiments that need to be performed.

I like the fact that authors did not overlook a simpler version of their idea, which was the Spectral mixture - LMC model. However, I do not think that the experiment they show in section 3 with results appearing in Figure 1 is completely fair. It is clear that a rank-1 SM LMC will not be able to give you different phases for the signals. Did you try different ranks for the corregionalization matrix? Did you try augmenting the value of Q for the SM-LMC kernel?

Clarity

Overall, the paper is clearly written. I think the DFT Approximation should have been included in the Materials and Methods section, and not in the Experiments section, perhaps in a similar fashion as you did for Section 3.

Some minor comments are as follows: - line 183. I think the work "logical" should be replaced with "natural". - The first reference in the References section is wrongly referenced. For starts, it was not published in the Machine Learning journal.

Originality

I think the paper attempts to fill a natural extension for the spectral mixture kernel, in the context of kernels for vector-valued functions.

Significance

The significance of the work can be improved at least in two ways, I think

- First, there is the issue of using the model for other type of signals. I think a more exhaustive experimental comparison is also missing. I did not see experiments for high-dimensional input spaces for example. I am also missing comparisons against the alternative models for kernels for vector-valued functions. - Second, the authors might do a better job in showing how important are their results in neuroscience. For the experiment in section 6.2, it might be good to show the raw data (or a part of it) so as to see why it is important to include the phase information or the dynamics of the cross-spectrum. It is also important to include quantitative results for the experiment in this section. Did you also try some other multi-output kernels on these data?
Summary: An interesting extension of the spectral mixture kernel to the multi-output GP framework. The significance of the work can be enhanced if more extensive experimental evaluation is included.

Submitted by Assigned_Reviewer_3

The topic of the paper is related to modeling of multi-output problems with GPs. In particular, the approach is applied to electrophysiological measurements from a mouse brain. Certainly the topic is important and interesting, and the application is quite timely.

The claimed contribution of the paper is to use extend a recently proposed mixture-form of GP kernel into multi-output one by using the LMC approach. To me this contribution seems pretty weak - it uses two well known methods to derive a single covariance kernel then uses it in a standard framework.

However, the application to HMM type of models is more of a contribution than the proposed kernel, but its novelty is not clear to me. It is not a claimed contribution so probably it is not novel then. The application to mouse brain data looks nice, but my knowledge on that application is not enough to judge whether this analysis is properly done and whether it really is compared to the state-of-the-art in that field. It is mentioned many times there though that it is just to "illustrate the utility of the CSM kernel in this application" so probably this means that the results are not really scientifically rigorous.
Summary: The paper is on modeling of multi-output problems with GPs with application to electrophysiological measurements from a mouse brain. The claimed main contribution is quite weak and it is hard for me to figure out the contributions of the other parts.

Author Feedback
Author rebuttal: We would like to thank the reviewers for the insightful comments. Our response is organized by topics in the order: contributions, results, inference, and references. We provide reviewer numbers (R1-R7) to indicate our response to specific comments.

* CONTRIBUTIONS (R3):
We do not claim the SM-LMC kernel as the main contribution of our work. Our main contributions are threefold: 1) we design a novel CSM kernel that elegantly encodes cross-power and cross-phase information into a multi-output covariance kernel; 2) we propose an efficient DFT method to fit the CSM kernel parameters; and 3) we demonstrate how the expressive capabilities of the CSM kernel provide rich brain state summaries of multi-channel LFPs.

* ADDITIONAL RESULTS (R2):
We hope to satiate concerns regarding 1) comparisons to alternative kernels for vector-valued functions, 2) quantitative results, and 3) experiments for different ranks of coregionalization matrices. The following discusses experimental results addressing these points.

We compare our CSM kernel, the SM-LMC kernel, and the SE-LMC (squared exponential) kernel. For each of these models we allow Q=20, and we vary the rank of the coregionalization matrices from rank-1 to rank-3. For a given rank, the CSM kernel always obtains the largest marginal likelihood for a window of LFP data, and the marginal likelihood always increases for increasing rank. To penalize the number of kernel parameters (e.g., a rank-3, Q=20 CSM kernel for 7 channels has 827 free parameters to optimize), we use the Akaike information criterion (AIC) for model selection (Akaike, "A new look at the statistical model identification," 1974). In the table below we show that a rank-2 CSM kernel is selected using this criterion, followed by a rank-1 CSM kernel. To show the rank-2 CSM kernel is consistently selected as the preferred model we report means and standard deviations of AIC value differences across 30 different randomly selected 3-second windows of LFP data.

rank | SE-LMC | SM-LMC | CSM
1 | 4770 (993) | 512 (190) | 109 (110)
2 | 5180 (1120) | 325 (167) | 0 (0)
3 | 5550 (1240) | 412 (184) | 204 (71.7)
Table 1: The mean and standard deviation of the difference between the AIC value of a given model and the AIC value of the rank-2 CSM model. Lower values are better.

We are also willing to display segments of the raw LFP data in the paper, as this will help to illustrate the presence of band-dependent phase shifts in LFPs. Finally, we believe comparisons to other types of signals and higher-dimension input spaces are interesting applications for future works. Although we agree they would improve the paper quality, we urge the reviewer to consider paper length constraints and the neuroscience focus of this work and the NIPS community.

* LINEAR CONJUGATE GRADIENTS(R1,R5):
In our applications, we have considered up to C=16 channels with N=600 sample locations. The full covariance matrix has undesirable O(N^2 C^2) storage and prohibitive O(N^3 C^3) inversion. With our method, we reduce the storage to O(N C^2) and inverse complexity to O(N C^3). Our inverse is not O(N logN C^3) because we use the functional form of the spectral density, negating the need to take the FFT of the first column of the circulant matrix. The reviewers are correct in pointing out that linear conjugate gradients could further reduce the complexity of this inverse. However, for the small C we consider (e.g., C=16), we have found that our FFT method is faster than conjugate gradients, and this is an important point to discuss further.

* VB INSTEAD OF EP/LAPLACE/MCMC (R5):
A Bayesian formulation of a HMM is convenient because it inherently penalizes for model complexity by imposing state transition constraints through Dirichlet priors. MCMC methods pose difficulties when evaluating convergence and estimating parameters; Laplace methods perform poorly with the sum-to-one constraints of the state transitions; and EP methods often have convergence issues. Variational inference is a straight-forward method to invert the Bayesian HMM. We recognize that VB underestimates the posterior variance, but this is not a major concern in our application. Inducing points were not necessary given our approximate inference technique.

* LOCAL MODES (R7):
Given large Q and using resilient back-propagation (rprop) for gradient-based optimization we find the SM kernel effectively fits a wide range of spectral densities without worries about local modes.

* REFERENCES (R1,R2):
We are more than happy to add the suggested, very appropriate references. We will reference Wilson and Adams (2013) in the abstract, reference and discuss Yang et. al (2015), and discuss the similarity between the the SM-LMC framework and the non-stationary proposal of a Gaussian process regression network (GPRN) in Wilson (2014). The first reference will be corrected to the journal "Foundations and Trends in Machine Learning" in 2012.